# Growth Light Quality Influences Leaf Surface Temperature by Regulating the Rate of Non-Photochemical Quenching Thermal Dissipation and Stomatal Conductance

**DOI:** 10.3390/ijms242316911

**Published:** 2023-11-29

**Authors:** Magdalena Trojak, Ernest Skowron

**Affiliations:** Department of Environmental Biology, Jan Kochanowski University of Kielce, Uniwersytecka 7, 25-406 Kielce, Poland; ernest.skowron@ujk.edu.pl

**Keywords:** leaf surface temperature, thermal imaging, chlorophyll fluorescence quenching, indoor farming, light quality, LED, non-photochemical quenching, FLIR

## Abstract

Significant efforts have been made to optimise spectrum quality in indoor farming to maximise artificial light utilisation and reduce water loss. For such an improvement, green (G) light supplementation to a red–blue (RB) background was successfully employed in our previous studies to restrict both non-photochemical quenching (NPQ) and stomatal conductance (*g*_s_). At the same time, however, the downregulation of NPQ and *g*_s_ had the opposite influence on leaf temperature (*T*_leaf_). Thus, to determine which factor plays the most prominent role in *T*_leaf_ regulation and whether such a response is temporal or permanent, we investigated the correlation between NPQ and *g*_s_ and, subsequently, *T*_leaf_. To this end, we analysed tomato plants (*Solanum lycopersicum* L. cv. Malinowy Ozarowski) grown solely under monochromatic LED lamps (435, 520, or 662 nm; 80 µmol m^−2^ s^−1^) or a mixed RGB spectrum (1:1:1; 180 µmol m^−2^ s^−1^) and simultaneously measured *g*_s_ and *T*_leaf_ with an infrared gas analyser and a thermocouple or an infrared thermal camera (FLIR) during thermal imaging analyses. The results showed that growth light quality significantly modifies *T*_leaf_ and that such a response is not temporal. Furthermore, we found that the actual adaxial leaf surface temperature of plants is more closely related to NPQ amplitude, while the temperature of the abaxial surface corresponds to *g*_s_.

## 1. Introduction

Plants, as sessile organisms, have evolved different defence responses to combat environmental stresses, such as drought, suboptimal temperature, and light stress. Activated regulatory mechanisms might, however, overlap in non-obvious ways, affecting plant physiology. As an example, non-photochemical thermal dissipation and stomatal conductance regulation can be considered. Stomatal conductance (*g*_s_) is a major regulator of carbon dioxide and water vapour exchange between the leaf interior and the surrounding atmosphere [1]. Thus, the rate of *g*_s_ is related to crop yield, and its regulation needs to fulfil the opposite role of minimising water loss while, at the same time, sustaining effective CO_2_ influx. Yet, the restriction of leaf *g*_s_ via chemical and/or hydraulic signals affects the rate of transpiration (*E*) and, consequently, the leaf surface temperature (*T*_leaf_) [2]. Additionally, the thermal dissipation process, called non-photochemical quenching (NPQ), which is crucial to minimising the risk of potential damage, is complex. On the one hand, the induction of NPQ acts as a safety valve, allowing excess light energy absorbed by photosynthetic pigments to be dissipated in the form of harmless heat [3]. On the other hand, while the induction of NPQ is rapid, its relaxation is a prolonged process, limiting photosynthesis, despite the fact that the light intensity might have already shifted towards the optimal dose [4]. Furthermore, as the up- and down-regulation of both processes, i.e., *g*_s_ and NPQ, influence *T*_leaf_ in opposite ways, it is important to determine which factor plays a more prominent role in its regulation. NPQ includes components with different mechanisms and characteristic times, such as energy-dependent (qE), zeaxanthin (Z)-dependent (qZ), photoinhibitory (qI), and state-transition-related (qT) components [5]. However, considering the aims of this study, we focus on the qZ trait, because the accumulation of Z facilitates the dissipation of excess energy via heat [6].

Previous research concluded that the relationship between *g*_s_ and NPQ-generated heat is close almost exclusively under strong actinic light [7]. Other studies [8] also proved a positive, nonlinear correlation with the extent of NPQ induction, with higher leaf temperatures being observed in response to increasing illumination. Another study [9] stated that despite the existence of a positive relationship between NPQ and leaf temperature change (Δ*T*_leaf_), the strength of its coupling varies. Those authors concluded that enhanced heat exchange managed with NPQ induction might be latent due to additional uncontrolled heat dissipation or the closure of stomata.

Furthermore, our current understanding of light-driven modification of leaf temperature is restricted to analyses concerning changes in incident light intensity [10]. In such cases, when plants are grown under unified light, the actual foliar temperature measured under changing light intensity is primarily related to stomatal aperture regulation rather than NPQ amplitude [11]. However, it is still unclear how different light qualities affect *T*_leaf_. Such a response, however, was expected, based on the results of previous studies [3,12], and we found that stomatal conductance and NPQ are related to the lighting spectra applied during plant growth in controlled-environment agriculture. It was noticed that the progressive replacement of R light by G light in the growth RB spectrum decreases stomatal dimensions, thus reducing *g*_s_ and *E* and, consequently, improving water-use efficiency [12] while decreasing evaporative cooling and presumably increasing foliar temperature [11]. Additionally, the influence of *T*_leaf_ on *g*_s_, which exists independently of the effects exerted through changes in plant water status [13], should be mentioned. Previous research [14] postulated that leaf hydraulic and mesophyll CO_2_ conductance, which are influenced by leaf anatomical traits (mesophyll type), are responsive to leaf temperature, especially in C3 species.

Moreover, the examined NPQ value in tomato plants grown under monochromatic (R, G, or B) or mixed RGB (1:1:1) light confirmed that R- and B-light treatments enhanced NPQ amplitude, while plants grown under G and RGB spectra presented a significantly lower amplitude of NPQ due to the reduced accumulation of NPQ-related proteins (PsbS, VDE, cyt*f*, and PGRL1) [3]. Another study [15] documented that NPQ amplitude increased in tomato plants grown under RB- and monochromatic B-light spectra compared to values noted under white light with G-light (544 nm) peak wavelength.

Taken together, the above results prompted questions regarding the role of the spectrum in determining leaf temperature. Firstly, the intriguing question is whether the application of monochromatic G light for plant cultivation or its introduction to the RB spectrum increases the actual *T*_leaf_ due to reduced *g*_s_ or decreases it due to the restricted induction and amplitude of NPQ. It might be also presumed that the mentioned effects exerted by specific light simply cancel each other out. Secondly, we asked if we would be able to notice differences in *T*_leaf_ related to the previous growth spectrum as a temporal or permanent phenomenon. In the first case, it could be explained by the redox state change or chloroplast inner membrane reorganisation and/or light-driven stomatal movement; thus, such a response might have been omitted from previous analyses. While the permanent phenomenon should be addressed with regard to the different patterns of gene expression and protein accumulation and/or modified leaf anatomy, it would be easier to notice, as it does not expire quickly during measurements when plants are exposed to certain actinic light or dark acclimation. Thus, the underlying concept of the present study is to elucidate the influence of spectrum-related thermal dissipation and stomatal conductance regulation on actual foliar temperature, leading to differences in evaporation and, consequently, water use efficiency in plants grown under different light regimes. Furthermore, as associated fluctuations in *T*_leaf_ under different light regimes are expected to influence photosynthesis directly, because it is a highly temperature-dependent process [10], we also analysed the photosynthetic activity of plants.

To this end, we employed different independent experiments with tomato plants grown under certain light regimes. First, we measured the *T*_leaf_ directly in growth chambers with a forward-looking infrared (FLIR) camera and analysed stomatal traits in the tomato plants. Then, we simultaneously measured the *g*_s_ and temperature of the stomata-rich abaxial leaf surface within an infrared gas analyser (IRGA) and a calibrated thermocouple placed inside the transparent gas exchange cuvette, as described previously [16], illuminated with RGB light at constant intensity. Next, we also analysed the gas exchange parameters within an IRGA, but in response to increasing light intensity. Then, based on methods described earlier [7,17], we conducted analyses of chlorophyll fluorescence imaging with a PAM fluorometer, monitoring NPQ amplitude with increasing light intensity (RLC—rapid light curve), and undertook a concurrent analysis of adaxial (illuminated) leaf surface temperature with a FLIR camera. Additionally, to distinguish between NPQ- and *g*_s_-related *T*_leaf_ change during RLC, we applied dithiothreitol (DTT) as a potent inhibitor of qZ component increase, thus significantly lowering the NPQ [3]. Then, we plotted a model curve of the obtained results of ΔNPQ/ΔNPQ_DTT_ or Δ*g*_s_ against Δ*T*_leaf_ and assessed the data fitting. This study provides valuable information about imaging methodologies, such as FLIR and PAM, that are suitable for predicting plant response to different light compositions in terms of relevant traits such as the rate of evaporation, photochemical light utilisation, and leaf temperature.

## 2. Results

### 2.1. Influence of Growth Light Spectra on Energy Quenching and Adaxial Leaf Surface Temperature

The effect of different growth light qualities (Figure 1) on the adaxial surface temperatures of tomato plant (*Solanum lycopersicum* L. cv. Malinowy Ozarowski) leaves 28 days after transplanting (DAT) was measured with a forward-looking infrared (FLIR) camera (FLIR E50) under a chamber-specific light composition and an air temperature of 22 ± 1 °C. We observed no significant differences in adaxial foliar temperature in plants grown under the combined RGB spectrum compared to the G or B groups, despite the higher PAR in the RGB chamber compared to those in the G or B one (180 µmol m^−2^ s^−1^ versus 80 µmol m^−2^ s^−1^) (Figure 2). Furthermore, an approximately 5% higher foliar temperature was noted under the R LEDs emitting photons of lower energy and longer wavelength than those of G and B LEDs. The results proved that under low light intensity (≤180 µmol m^−2^ s^−1^), differences in light quality are of minor importance in the regulation of foliar temperature of the adaxial side.

To elucidate plant response to higher PAR, we further monitored the temperature of the abaxial side of the leaf under such conditions. To overcome the possible limitation of Z synthesis during NPQ formation, plants of all groups were exposed to 30 min pre-illumination with RGB light (400 µmol m^−2^ s^−1^). Meanwhile, to inhibit the production of violaxanthin de-epoxidase enzyme (VDE), a part of the leaves was infiltrated with DTT. The influence of DTT on stomatal conductance was also analysed during gas exchange measurements.

The highest value of Fv/Fm of the water-treated sample was presented by plants grown under monochromatic blue light (Fv/Fm = 0.745) and the lowest under red (Fv/Fm = 0.547) and green (Fv/Fm = 0.599) light (Figure 3). At the same time, under the mixed RGB spectrum, we observed Fv/Fm = 0.717, which confirmed the postulated positive influence of B light on PSII vitality. We noted that DTT infiltration decreased Fv/Fm by about 14, 3, and 3% for RGB, G, and B light, respectively, while in plants grown solely under R light, DTT treatment exerted a positive influence on Fv/Fm (+14%). The observed effect of DTT infiltration was more pronounced compared to previous research due to the enhanced infiltration period (60 min versus 30 min) and the application of whole leaves immersed with petioles in a DTT-filled tube instead of a leaf disc floating on a DTT solution. Nonetheless, the observed Fv/Fm values, in all treatments, were lower than expected, indicating stressful conditions. A likely explanation for this is that the leaf samples of plants grown under low PAR were subsequently exposed to higher light intensity during ChlF measurements, thus lowering the Fv/Fm value.

We also analysed the induction of NPQ during an RLC assay under increasing light intensity (0–1250 µmol m^−2^ s^−1^) with simultaneous thermal imaging of *T*_adaxial_ at the plateau phase of NPQ induction at 396 µmol m^−2^ s^−1^ of water- (Figure 4a) and DTT-infiltrated leaves (Figure 4b). The analysed NPQ kinetics revealed a significant difference in maximal amplitude between groups, both at plateau PAR as well as at the PAR endpoint (1250 µmol m^−2^ s^−1^). In the water-treated samples, the highest NPQ level was observed in B and R plants, reaching 1.0 and about 0.75 (at 396 µmol m^−2^ s^−1^), respectively, whereas in the RGB and G groups at the plateau level, NPQ reached 0.57 and 0.33, respectively. Moreover, the corresponding abaxial surface temperature reached 31.5, 29.2, 28.7, and 27.1 °C in B-, R-, RGB-, and G-light-grown plants, respectively. As expected, in the case of leaves infiltrated with DTT, we noted significantly reduced NPQ amplitudes of 72, 60, 42, and 50% in RGB, R, G, and B plants, respectively, compared to the water control, whereas the corresponding *T*_adaxial_ decreased by approximately 7, 6, 1, and 6% in B, R, G, and RGB plants, respectively.

It should, however, be noted that ChlF analyses were performed at 25 °C; therefore, to estimate the proper relationship between qZ NPQ induction and concomitant foliar temperature change during RLC, we plotted ΔNPQ against Δ*T*_adaxial_, which was assessed by subtracting the NPQ or adaxial leaf surface temperature (FLIR) values estimated for the control and DTT-infiltrated leaves at the same point of light intensity and quality (396 µmol m^−2^ s^−1^, 450 nm) (Figure 5). The results showed a clear correlation between the inhibition of the DTT-sensitive NPQ component and the restriction of temperature change in the upper leaf side, with both the linear fitting model and the non-linear Bradley model presenting with a 95% confidence band. Both functions presented satisfyingly high goodness of data fitting, with adjusted R^2^ of 0.997 and 0.996. We tested linear and non-linear regression models. The highest foliar temperature of the upper side accompanied by an induction of qZ-dependent heat dissipation was noted for plants previously grown under monochromatic B light, followed by the R and RGB plants, while the lowest values were recorded for G light-grown plants. The data obtained for all groups produced linear functions. Thus, the results proved the existence of a relationship between heat dissipation through qZ induction and an increase in the foliar temperature of the adaxial, illuminated leaf side at higher PAR. Furthermore, as expected, we observed differences in NPQ amplitude and *T*_adaxial_ as a result of previous growth light conditions and noticed that such a response was rather a permanent feature and did not dissipate after pre-illumination and dark adaptation.

### 2.2. Influence of Growth Light Spectra on Stomatal Traits, Gas Exchange, and Abaxial Leaf Surface Temperature

The results showed that green light, applied solely for tomato cultivation, influences the distribution of the stomata in the abaxial epidermis (Figure 6).

As expected, the lowest *S*_d_ was observed in plants which developed in the presence of G light, while the highest occurred in those that developed under R and B light (Table 1). The pore area per leaf area was also found to be the highest under R light treatment and the lowest for G light, indicating that the effective area of transpiration within G plants was significantly reduced. This was attributed to both *S*_d_ and the pore width of individual stomata under G treatment (Figure 7, Table 1) compared to other light treatments. As a result, stomatal pores were reduced under G treatment by approximately 50% compared to RGB. At the same time, we documented that the area of the stomatal complex was positively correlated with light intensity and not related to the applied light quality. Interestingly, the analysed abaxial epidermal layers showed curious phenomena of trichome patterns, presenting unequal distribution under different light treatments (Figure 6) which might have interfered with the transpiration rate and, consequently, modulated the foliar temperature. This requires further investigation in the future. It should, however, be noted that despite the R plants showing a high pore area per leaf area parameter, the overall area of the abaxial leaf surface was reduced under R light as a consequence of the development of smaller leaves. Consequently, the total pore area per total abaxial leaf surface was the highest under RGB and B lights and was reduced by 34 and 76% in R and G plants, respectively (Table 1).

Analyses of stomatal traits provided a better understanding of recorded differences in gas exchange and abaxial side leaf temperature among groups. Analyses were performed with water- and DTT-infiltrated leaves illuminated under constant PAR = 400 µmol m^−2^ s^−1^, corresponding to the conditions of NPQ and *T*_adaxial_ measurements. The results of the net photosynthetic rate (*P*_n_) (Figure 8a) showed similar tendencies to the Fv/Fm values (Figure 3). Plants grown under monochromatic R and G light presented reduced CO_2_ fixation (under RBG light illumination) by 24 and 62% compared to RGB plants, respectively. At the same time, plants grown under monochromatic B light sustained RGB-like levels of CO_2_ fixation. We also observed that DTT treatment increased *P*_n_ in R but decreased it in RGB plants, due to its influence on stomatal conductance (Figure 8b). The analysed *g*_s_ showed that DTT reduced stomatal conductance by 63, 43, and 29% in RGB, G, and B plants, respectively. Consequently, the transpiration rate (Figure 8c) was also reduced after DTT application in these groups. In contrast, in R plants, DTT enhanced stomatal conductance and transpiration by 24 and 33%, respectively. Moreover, as expected, the temperatures of the non-illuminated abaxial leaf side measured under 400 µmol m^−2^ s^−1^ were lower (Figure 8d) than those of the adaxial side (Figure 4a), which might be considered a consequence of the absence of direct light exposure and the cooling effect of transpiration.

Nonetheless, a comparison of the *g*_s_ and *T*_abaxial_ values in water- and DTT-infiltrated samples of RGB, R, and B plants indicated a sort of positive relationship. The curve plotted in Figure 9, obtained by fitting the experimental data of *g*_s_ and *T*_abaxial_, presents satisfyingly high goodness of data fitting, with an adjusted R^2^ of 0.984, thus providing direct evidence of a close connection between *g*_s_ and *T*_abaxial_.

### 2.3. Influence of Growth Light Spectra on Gas Exchange and Abaxial Leaf Surface Temperature in Response to Increasing Light Intensity (LC)

An analysis of *T*_abaxial_ and *g*_s_ at a constant light intensity showed a notable relationship between the two variables (Figure 9). Thus, in the next experiment, we assessed the relationship between foliar temperature and stomatal conductance in response to increasing light intensity to determine the light intensity above which the cooling effects of transpiration became limited. To overcome the limitation of stomatal conductance, we applied the procedure of a rapid light response curve with pre-illuminated leaves and a stepwise decrease in light intensity (LC). The drawback of such an assay is an overestimation of stomata openness compared to the measurement at constant PAR without previous pre-illumination. However, the primary reason to conduct such an experiment was to estimate the threshold level of PAR above which the close relationship between *g*_s_ and foliar temperature regulation tended to fail. The results of the gas exchange measurements showed that the rate of net photosynthesis (Figure 10a), stomatal conductance (Figure 10b), and transpiration (Figure 10c) were the highest in B plants and the lowest in plants grown under G light. To plot curves, we employed regression models fitting experimental data and showed that *P*_n_ reached a plateau at PAR ≥ 800 µmol m^−2^ s^−1^, while *g*_s_ reached a plateau at PAR ≥ 400 µmol m^−2^ s^−1^ in all light treatments. At the same time, the transpiration rate (Figure 10c) and abaxial leaf temperature (Figure 10d) increased continuously in response to PAR increase. Thus, we observed a steady increase in abaxial side temperature, despite the concomitant increase in transpiration, indicating that the cooling effects of transpiration at higher PAR were limited.

**Figure 10 ijms-24-16911-f010:**
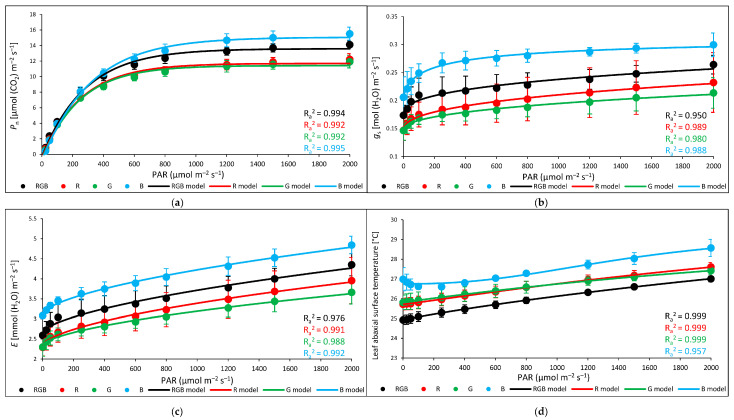
Gas exchange parameters of control leaves of tomato plants (*Solanum lycopersicum* L. cv. Malinowy Ozarowski) 28 DAT, measured with an LED light source 6400-02B, illuminated with 0–2000 µmol m^−2^ s^−1^ of RB light (R:B 10:1). (**a**) Net photosynthetic rate (*P*_n_), (**b**) stomatal conductance (*g*_s_), (**c**) transpiration rate (*E*), and (**d**) leaf abaxial surface temperature. Each data point represents the average ± SD of six independent measurements (*n* = 6). We employed (**a**) an asymptotic regression model, (**b**,**c**) a lognormal cumulative distribution function, and (**d**) a logistic regression model to fit the experimental data (points; black—RGB, red—R, green—G, blue—B chamber) of gas exchange parameters against PAR. Fitting was applied (as specified in Section 4) and is reported with adjusted R^2^ (R_a_^2^) values, used to determine the goodness of data fitting.Also, the analysed water-use efficiency (WUE), which expresses the trade-off between carbon assimilated as biomass or grain and units of water release [18], showed that the efficacy of water utilisation increased up to approximately 800 µmol m^−2^ s^−1^ of PAR. The results showed that for all light treatments, both intrinsic water-use efficiency (*P*_n_/*g*_s_) (*WUE*_int_) (Figure 11a) and instantaneous water-use efficiency (*P*_n_/*E*) (*WUE*_ins_) (Figure 11b) increased rapidly under lower light intensity before reaching the maximum value of WUE. In the case of *WUE*_ins_, the lowest water-use efficiency was noted for B plants, but without statistically significant differences recorded among groups, while when *WUE*_int_ was analysed, B plants presented decreased water-use efficiency in response to 200–1000 µmol m^−2^ s^−1^ compared to other plants.

**Figure 11 ijms-24-16911-f011:**
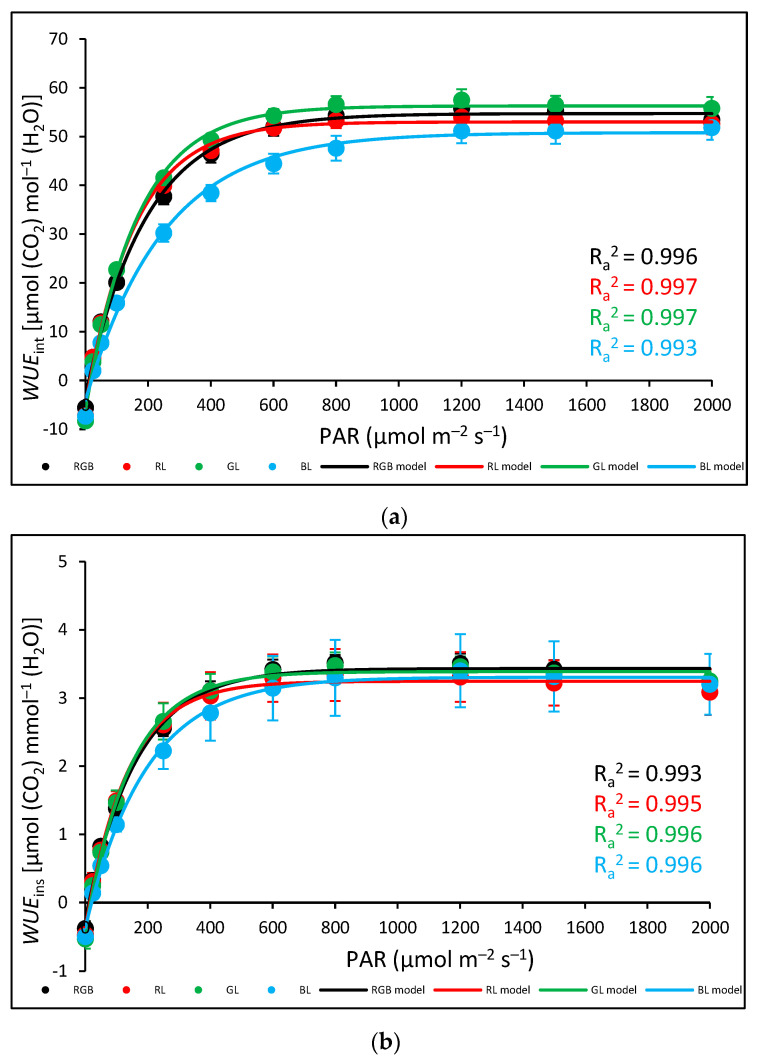
Gas exchange parameters of control leaves of tomato plants (*Solanum lycopersicum* L. cv. Malinowy Ozarowski) 28 DAT, measured with LED light source 6400-02B, illuminated with 0–2000 µmol m^−2^ s^−1^ of RB light (R:B 10:1). (**a**) Intrinsic water-use efficiency (*P*_n_/*g*_s_) (*WUE*_int_), (**b**) instantaneous water-use efficiency (*P*_n_/*E*) (*WUE*_ins_). Each data point represents the average ± SD of six independent measurements (*n* = 6). We employed (**a**) an asymptotic regression model and (**b**) a sigmoidal model to fit the experimental data (points; black—RGB, red—R, green—G, blue—B chamber) of gas exchange parameters against PAR. Fitting was applied (as specified in the Section 4) and is reported with adjusted R^2^ (R_a_^2^) values, used to determine the goodness of data fitting.

The threshold level of the influence of *g*_s_ on the leaf abaxial surface temperature is presented in Figure 12. Curves plotted with experimental data fitting for each chamber revealed that, while the mechanism of stomatal openness effectively stabilised abaxial temperature at lower PAR, it started to fail above approximately 400 µmol m^−2^ s^−1^ (encircled data points), despite the continuously rising rate of transpiration.

## 3. Discussion

It has been previously reported [19] that long-term light-quality treatment has an impact on photosynthetic properties in rice. The authors of that report showed that even under moderate incident light levels (300 µmol m^−2^ s^−1^), monochromatic red and even more blue light enhanced NPQ induction compared to white light treatment. Another study [20] concluded that NPQ, which is considered important for short-term light acclimation, is also an important mechanism for long-term light acclimation. Similarly, in our previous research [3], we documented that both long-term R- and B-light treatment increased NPQ amplitude, while G and RGB light lowered its value due to the reduced accumulation of NPQ-related proteins. At the same time, previous research [12,21,22,23,24] has shown that the quality of growth light influences gas exchange, especially due to the modification of stomatal development and movement. The main conclusions from these studies are that B light stimulates stomata opening more than R light, while the addition of G light to the spectrum reduces stomatal conductance via both contractions of stomata dimension and density, as well as reduced stomatal openness. Thus, firstly, the influence of growth light composition on NPQ and *g*_s_ has been well established; secondly, the effect of these mechanisms on foliar temperature has also been studied [7,8,11,25,26]. Nonetheless, relatively little research has focused on the direct influence of NPQ and *g*_s_ on the leaf surface temperature due to the different spectral characteristics applied during plant cultivation. In this regard, it is crucially important to identify the mechanism that overrides foliar temperature regulation, to try to evaluate its limits, and to link it to the long-term exposure of plants to the growth spectrum.

### 3.1. Effect of Long-Term Exposure of Tomato Plants to Different Light Compositions on Maximum PSII Photochemical Efficiency

In the present study, we have demonstrated that growth light composition influences the subsequent maximum PSII photochemical efficiency Fv/Fm, the amplitude of non-photochemical quenching, and, consequently, the adaxial surface temperature of leaf lamina. It has been previously documented that for unstressed leaves, the Fv/Fm value is highly consistent and oscillates at about a 0.83 value, correlating to the maximum yield of photosynthesis [27]. In agreement with previous reports [3,28], we observed that B light, applied both as an exclusive light source or in combination with R and G light, increased Fv/Fm. At the same time, the results showed that monochromatic R light exerted the most adverse effects on Fv/Fm. Another study [29] noted that the photosynthetic apparatus of potato plantlets under monochromatic R light was impaired compared to those grown under monochromatic B or combined RB spectrum. In addition, the authors of [30,31] reported the adverse effects of monochromatic R light on Fv/Fm. Such a phenomenon is called “red-light syndrome”, the existence of which is supported by the fact that it is evoked when plants are grown solely under R light [32]. Symptoms of this syndrome include a strong decrease in photosynthetic capacity and Fv/Fm, as well as unresponsive stomata. The observed response is related to the reduced number of chloroplasts and thinner layers of mesophyll tissues, resulting in lower photosynthetic capacity. Interestingly, in our research, R-light-grown plants also displayed some morphological features related to red-light syndrome, such as leaf curling. Furthermore, we observed that only in R plants did the application of DTT increase Fv/Fm, while in other plants, it lowered its value. Similar tendencies have been documented in our previous research [3]. The authors of [33] stated that DTT treatment might shift the balance of excitation energy distribution between PSI and PSII, increasing energy partitioning to PSII. Additionally, research [34] has shown that DTT, as a reductant agent, interferes with the chloroplast redox state, thus reducing the photoprotective energy spillover from PSII to PSI, which results in a higher Fv/Fm value. The relationship between long-term acclimation to spectrum composition and energy balance between PSII and PSI is related to the major light-harvesting complex (LHCII) association with each photosystem [35]. The association of most of the LHCIIs with PSII causes an increased absorption of light in the blue region of the spectrum by PSII compared to R light, while the partial migration of LHCIIs to PSI allows for a more equal absorption in the blue and red spectrum with both photosystems [36]. Overall, we concluded that the application of 435 nm of B light, as a sole light source or RGB component, exerted a positive influence on photochemical capacity due to more efficient energy partitioning between PSII and PSI, while the opposite trend was observed for monochromatic R light at 662 nm. In the case of 520 nm monochromatic G light, the observed reduction in the Fv/Fm value was related to reduced chlorophyll content, as has been documented previously [3,37].

### 3.2. Effect of Long-Term Exposure of Tomato Plants to Different Light Compositions on Energy Quenching and Adaxial Leaf Surface Temperature

Contrary to our expectation, we found no differences in the foliar temperature of the upper leaf side when measured directly inside chambers (in situ), despite the difference in the photon flux and spectral characteristics among RGB and G or B lights. At the same time, we noted on average 1.0 °C higher values within the plants grown under monochromatic R light. Overall, foliar temperature, measured in situ in all light treatments, was found to be within a range of 1.5–2.5 °C below the air temperature. According to a previous analysis [38], if plants are not water stressed, the radiation source exerts a much smaller effect than the plant water status or leaf evaporative cooling. However, when the cooling mechanism decreases or becomes inefficient, the leaf temperature can increase well above the ambient temperature [38].

Thus, in the next experiment, we evaluated the foliar temperature change in response to increasing photon flux. Our previous research showed that long-term exposure to G light compared to B or R light significantly reduces the accumulation of NPQ-related protein and, consequently, its amplitude and heat dissipation [3]. A previous study [39] stated that terrestrial plants are rather fine-tuned to reduce energy absorption and have developed excessive pigments such as carotenoids and anthocyanins that effectively reduce the flux of mostly shortwave (including B light) PAR, containing much of the surplus energy. At the same time, the R and B light photons are absorbed more strongly by photosynthetic pigments than G photons; thus, their absorption near the adaxial surface also increases heat dissipation [40]. In such a scenario, it is expected that in a high light intensity background, additional R or B light will cause the up-regulation of NPQ-related heat dissipation and increased *T*_leaf_ [41].

Based on these assumptions, we decided to test whether the spectrum composition influenced NPQ amplitude when analysed in response to increasing PAR and, consequently, whether it would affect the upper leaf side temperature. To this end, we applied blue actinic light during the NPQ measurements of plants grown under different spectra and observed differences among groups, as well as the positive correlation with the extent of NPQ- and FLIR-recorded adaxial leaf surface temperature (Figure 4a). The highest *T*_adaxial_ in the control was noted for the B-light tomato plants, which consequently presented the highest NPQ amplitude, while the lowest values were noted for the G group. Furthermore, this positive correlation between spectrum-related *T*_adaxial_ and NPQ values was also proven with a subsequent analyses of DTT-infiltrated leaves, indicating that reduced NPQ interfered with lowered leaf temperature (Figure 4b). Previously, the authors of [8] also reported the correlation between the extent of NPQ and light-induced temperature increase in tobacco leaves. In contrast, other research [11] has documented no direct correlation between the PsbS-related NPQ amplitude and foliar temperature dynamics. Such a conclusion, however, might be a sort of oversimplification, as the authors of [41] identified two independent quenching sites depending on PsbS or zeaxanthin, while the authors of other studies [42,43,44,45] documented that both PsbS-deficient (*npq4*) and zeaxanthin-deficient (*npq1*) Arabidopsis plants showed a similar reduction in the maximum NPQ level compared to wild-type plants. Another feasible method of dissipating absorbed energy, such as carotenoid radical cation formation (qZ), was also postulated [11]. According to the authors of [42], both PsbS and Z are crucial elements of photoprotective energy dissipation, but their roles are different. PsbS proteins interfere with the formation of densely packed aggregates of thylakoid membrane proteins, thus allowing the exchange and incorporation of xanthophylls. At the same time, zeaxanthin formation in place of V with VDE was shown to enhance direct excitation quenching.

Based on these results, the applied DTT allowed for the inhibition of VDE activity and proved the influence of a more slowly inducible zeaxanthin-dependent component of non-photochemical quenching on the subsequent foliar temperature, with both linear and non-linear fitting models presenting satisfyingly high goodness of data fitting (Figure 5). We tested both models, as previous research [11] postulated that at lower light intensities, this relationship is non-linear, while it becomes linear at higher light intensities. Moreover, analyses of NPQ and *T*_adaxial_ in DTT-infiltrated leaf, as well as our previous analyses of PsbS and VDE levels under different spectra [3], proved that the relationship between PsbS-dependent NPQ amplitude and *T*_leaf_ regulation is minor when Z accumulation is inhibited (Figure 4b).

### 3.3. Effect of Long-Term Exposure of Tomato Plants to Different Light Compositions on Gas Exchange Parameters, Stomatal Traits, and Abaxial Leaf Surface Temperature

At the same time, the role of stomatal conductance in leaf energy balance cannot be overlooked. However, as previous analyses [46] indicated that stomata in the leaves of tomato plants are predominantly placed on the abaxial side, we recorded the relationship between *g*_s_ and *T*_abaxial_ with an IRGA and a thermocouple, respectively. Such an approach has advantages, as it allowed us to measure the straightforward influence of light-regulated stomatal openness on abaxial leaf side temperature and minimise the effect of direct illumination on temperature change. In a previous study [11], the authors concluded that foliar temperature dynamics are primarily affected by stomatal aperture in response to intense irradiation. Those authors, however, estimated stomatal aperture indirectly, i.e., with a crop water stress index (CWSI) indicator, while the FLIR-recorded *T*_leaf_ was monitored for the adaxial side, which is less occupied by stomata. Unfortunately, the CWSI indicator may have overestimated *g*_s_; thus, a new thermal indicator of stomatal conductance (GsI), as a more reliable tool for indirect stomatal conductance estimation, was proposed [1]. In our research, we documented negative, non-linear correlations among directly and simultaneously recorded *T*_abaxial_ and *g*_s_ under constant PAR (400 µmol m^−2^ s^−1^) (Figure 9), presenting satisfyingly high goodness of data fitting with adjusted R^2^. However, when the data were analysed with increasing light intensity, this correlation became less obvious (Figure 10b–d). It was found that above the 400 µmol m^−2^ s^−1^ of PAR, stomata seemed to approach their maximal aperture and the cooling effect, which stabilised the abaxial side temperature in lower PAR (Figure 8), tended to fail (Figure 12), although the transpiration rate still increased (Figure 10c). Moreover, the recorded negative non-linear correlation between *g*_s_ and *T*_abaxial_ under constant PAR (Figure 9) was also disturbed when analysed within the light response curve (LC). We found that within LC, the highest *T*_abaxial_ was noted for B-light-grown plants (Figure 10d), presenting both the highest *g*_s_ and transpiration rate (Figure 10b,c), while the lowest *T*_abaxial_ was presented in RGB plants. Additionally, the analysed WUE parameters (Figure 11) confirmed that the enhanced *P*_n_ during LC noted for the B plants, which resulted from the higher stomatal conductance, reduced the efficiency of water use per unit of fixed carbon dioxide. Interestingly, comparing the influence of the spectrum on the *g*_s_ noted under constant PAR (Figure 8b) and LC (Figure 10b), we found that the stomata of B-light-grown plants presented enhanced responsiveness to increasing light intensity compared to RGB; this cannot be simply explained by the differences in stomatal traits. Consequently, based on the analysed stomatal traits (Table 1), we concluded that the size of the stomata complex is related to the applied light intensity, while the stomatal density and pore area are affected by light quality. Both monochromatic R and B light enhanced the stomatal density (*S*_d_). At the same time, however, the stomata of R plants remained significantly less sensitive to higher PAR; such a response has been postulated to be a result of red-light syndrome [32]. Furthermore, we observed that the long-term exposure of tomato to monochromatic G light caused a reduction in stomatal aperture within the pore width, while G and G-enriched spectra significantly lowered the number of stomata per abaxial leaf area. Additionally, our previous analysis [12] proved that the replacement of a red light with a green one induces a contraction of stomatal dimensions. It was especially noted that increasing G light intensity reduced pore area by nearly one-third, reduced *S*_d_ when occupying 20–40% of the spectrum, and significantly reduced the overall pore area estimated per leaf area. Consistent with our data, previous studies [30] also documented that B light facilitates stomatal opening and density compared with G-light treatment. Moreover, as G light reverses the effect exerted by B light, its addition to the RB spectrum could enhance drought tolerance by altering the stomatal aperture [47] and reducing stomatal density, as documented for the RGB treatment in this study. The disadvantages of the application of monochromatic G-light for plant cultivation are reduced *P*_n_ and increased *T*_abaxial_, resulting from significantly lower stomatal conductance. It has been shown [30] that monochromatic light of each kind modifies the hormonal balance within the leaf. The authors concluded that the mechanism by which B light regulates *g*_s_ is related to decreased abscisic acid (ABA) levels under B light, whereas G light enhanced ABA level and ABA-sensitivity due to the up-regulation of ABA-responsive element-binding proteins (*AREB1*) [47].

## 4. Materials and Methods

### 4.1. Plant Material and Growth Conditions

Tomato (*S. lycopersicum* L. cv. Malinowy Ozarowski) seeds were germinated in Petri dishes on sterile filter papers soaked in Milli-Q water at 26 °C. For analysis, a tomato cultivar with reduced leaf dissection (potato leaf phenotype) [48] was chosen to maximise light absorption in the upper part of the canopy. The two-week-old seedlings were transplanted to P9 containers (9 × 9 × 10 cm) and filled with a substrate (white and black peat, perlite, and N:P:K = 9:5:10; pH, 6.0–6.5), divided into groups, and transferred to environmentally controlled growth chambers with non-reflective black separators to eliminate light contamination. The plants were grown for the next 28 consecutive days (28 days after transplanting (DAT)) under LED RhenacM12 lamps (PXM, Podleze, Poland) delivering 180 µmol m^−2^ s^−1^ of the RGB spectrum (R:G:B = 1:1:1) or 80 µmol m^−2^ s^−1^ of each monochromatic R, G, or B light (Figure 1). LED characteristics were as follows. Red LEDs: peak wavelength of 662 nm, peak broadness at half peak height of 22 nm (649–671 nm); green LEDs: 520, 34, 505–539 nm, respectively; and blue LEDs: 435, 17, 426–443 nm, respectively. RGB treatment was used as the control group. Light composition and photosynthetic photon flux density (PPFD) were monitored daily via a calibrated spectroradiometer GL SPECTIS 5.0 Touch (GL Optic Lichtmesstechnik GmbH, Weilheim/Teck, Germany). The readings were averaged for six locations at the level of the apical bud and maintained by adjusting the distance between the light sources and the plant canopy. The containers with tomato plants cultivated under the same light treatment were turned twice a day. To avoid canopy shading and overlapping, five plants per square meter of the illuminated area were cultivated. The photoperiod was 16/8 h (day/night; day 6:00 a.m.–10:00 p.m.), the average air temperature was maintained at 22/20 °C (day/night), and the relative air humidity was kept at 50–60% with 410 ± 10 µmol mol^−1^ of CO_2_. The plants were watered with tap water when necessary and fertilised once a week with 1% (*w*/*v*) tomato fertiliser (N:P:K = 9:9:27; Substral Scotts, Warszawa, Poland). The fourth leaf from the above plants 28 DAT was used for subsequent analyses. All analyses were conducted between 8:00 a.m. and 12:00 p.m. Ten tomato plants (two samples with five plants per light condition) were grown under each kind of light treatment.

### 4.2. Pre-Illumination of Dithiothreitol-Infiltrated Leaf Samples

DTT (Sigma-Aldrich, St. Louis, MO, USA) pre-treatment of leaves was used to inhibit the VDE that promotes the qZ component of NPQ induction, related to V de-epoxidation to Z [3]. At the same time, DTT does not interfere with the development of the trans-thylakoidal ΔpH regulating qE, related to PsbS protein [34], which has been previously analysed [11].

Fully developed, detached leaves of tomato plants 28 DAT were placed immediately into 2 mL plastic tubes with a hole in the lid for the leaf petiole. These tubes contained 5 mM DTT solution or distilled water (C, control) [49] and were kept for 1 h in the dark. To avoid light contamination, leaves were cut off directly under the lighting conditions of the chambers.

After infiltration, C and DTT leaves of each chamber were pre-illuminated for 30 min with RGB light (R (627 nm):G (530 nm):B (447 nm) = 1:1:1) at an intensity of approximately 400 ± 5 µmol m^−2^ s^−1^ (LED Light Source SL-3500 lamp, Photon Systems Instruments, Drasov, Czech Republic). Such an approach allowed us to overcome the limitation of Z synthesis during NPQ formation due to activated VDE, which results in the accumulation of Z [42]. The light intensity and exposure periods applied for pre-illumination were chosen based on previous analyses [3,42,43,50] to allow for the acceleration of NPQ formation and to avoid PSII photoinhibition. Subsequently, leaves were re-darkened for 15 min to allow relaxation of the transthylakoid pH gradient to occur without substantial reconversion of Z back to V.

### 4.3. Rapid Light Curve (RLC) of Chlorophyll Fluorescence (ChF) Analyses

ChlF was measured using a pulse-amplitude-modulated (PAM) fluorometer (Maxi IMAGING PAM M-Series, Walz, Effeltrich, Germany) on the adaxial side of the leaf samples placed in plastic tubes. To avoid heterogeneity of DTT infiltration, six circle-shaped areas of interest (AOIs) were selected and averaged for each replicate. The minimal (dark) fluorescence level (Fo) was measured using measuring modulated blue light (450 nm, 0.01 µmol m^−2^ s^−1^). The maximal fluorescence level (Fm) with all PSII reaction centres closed was determined via a 0.8 s saturating blue light pulse (SP = 450 nm) at 5000 µmol m^−2^ s^−1^ in 30 min dark-adapted samples. The maximum PSII photochemical efficiency (Fv/Fm) was derived from that (Fv/Fm = (Fm − Fo)/Fm) value. Then, for our quenching analysis, leaf samples were illuminated for approximately 5 min at increasing blue actinic light intensity (AL = 450 nm) at the following steps: 0, 20, 55, 110, 185, 280, 335, 395, 460, 530, 610, 700, 925, and 1250 μmol m^−2^ s^−1^ [51]. Increasing the AL intensity during the measurement of C- and DTT-fed leaves was done to clarify the role of qZ and qE in the *T*_leaf_ regulation of plants grown under different light regimes while illuminating with excess energy excitation. To determine the NPQ induction, SPs (5000 µmol m^−2^ s^−1^, duration 0.8 s) were applied every 20 s for each AL intensity. To avoid qI induction, the RLC assay was shortened to 5 min. The fluorescence yields obtained during our analyses were combined to calculate the Stern–Volmer NPQ = (Fm − Fm’)/Fm’ (where Fm’ is the maximal level of chlorophyll fluorescence in light) [52]. The assessed NPQ value exceeded unity; thus, its value is presented as NPQ/4. All measurements of ChlF were carried out at a constant ambient temperature of 25 ± 1 °C. Each measurement comprised six replicates.

### 4.4. Determination of Leaf Temperature with FLIR

The temperature of the leaf surface was determined, as previously described [7,8,11], with FLIR E50 camera software (Systems, Wilsonville, OR, USA, https://www.flir.com/support/products/e50/#Overview, accessed on 12 October 2023), which detects temperature with a relative resolution of 0.05 °C based on the measurement of infrared radiation at 7.5–13 µm. The camera operates on the principle of object scanning and provides a spatial resolution of 320 × 240-pixel; images are recorded to a disk in the thermal camera. The pictures were captured with the maximal frequency of image refreshing (60 Hz). For analyses, leaves of similar size, age, and shape from each light regime were chosen. Infrared imaging of *T*_leaf_ (*T*_adaxial_) was performed on the adaxial leaf surface, either under the chamber-specific light and air temperature (at 22 ± 1 °C) conditions or during chlorophyll fluorescence imaging (at 25 ± 1 °C) after the NPQ parameter had reached a plateau phase of induction at 396 µmol m^−2^ s^−1^, according to previous results [7]. Each measurement comprised six replicates.

### 4.5. Stomatal Traits in the Various Lighting Spectra

Previous research [12] concluded that in response to increasing G-light intensity in the spectrum, plants tended to contract their stomatal dimensions and, consequently, exhibited reduced *g*_s_. Thus, in the present study, we analysed plants grown solely under monochromatic lights (R, G, and B) or mixed RGB to determine which applied light exerted the most pronounced effects on the stomata.

Leaf epidermal strips from plants 28 DAT were used to assess all morphological stomatal features. Ten randomly selected leaves (one leaf per plant) per light treatment were collected in the morning, and epidermal strips (four strips per leaf) were peeled off from the abaxial side of the leaf (avoiding leaf veins) and allowed to float on 2 mL of a basal reaction mixture (5 mM of 2-(N-morpholino)ethanesulfonic acid (MES), 50 mM of KCl, 0.1 mM of CaCl_2_; pH, 6.5) for 2 h [53]. Stomatal traits were analysed with the images obtained using a Nikon Eclipse E100 microscope with AxioVision 4.8 software (Carl Zeiss Inc., Oberkochen, Germany). A magnification of ×1000 was used to determine the individual stomatal traits, and ten randomly selected stomata per sampling area were measured. The stomatal complex width, length, pore width (minor axis of the pore), pore length (major axis of the pore), and stomatal width/length ratio were measured. For the width/length ratio, stomata width, including pore width, was chosen instead of the guard cell width, since the latter changes as the stomata close. The stomatal and stomatal pore area (µm^2^) were also determined using AxioVision 4.8. Stomatal density (*S*_d_) was determined under a magnification of ×400 with five different fields of view per sampling area. Pore area per leaf area was calculated as the pore area per stomata × *S*_d_, and total leaf pore area as the cumulative pore area of the stomata (based on *S*_d_) to the total abaxial leaf surface [22,54].

### 4.6. Leaf Gas Exchange and Temperature Determination of Plants Grown under Various Lighting Spectra

The photosynthetic parameters were measured using a Li-6400XT Portable Photosynthesis System (LI-COR Inc., Lincoln, NE, USA) in 2 × 3 cm transparent chambers (6400-08) that were illuminated directly with a RGB chamber-like light composition or with a 2 × 3 cm LED light source 6400-02B (665 ± 10 nm and 470 ± 10 nm) to obtain the light response curves (LCs). The leaf cuvette conditions were maintained at a relative air humidity of 60%, 400 µmol mol^−1^ of external CO_2_ concentration, gas flow rate of 500 ± 2 µmol s^−1^, and a block temperature set to a constant 25 °C. Analyses with the transparent gas chamber were conducted under 400 µmol m^−2^ s^−1^ of the RGB spectrum (R:G:B = 1:1:1), with 300 s per replicate to stabilise the gas exchange. For the LC analysis, we applied a rapid light response curve procedure with pre-illuminated leaves and a stepwise decrease in light intensity at the following steps: 2000, 1500, 1200, 800, 600, 400, 250, 100, 50, 25, and 0 m^−2^ s^−1^. During the LC, steps were time-separated (200 s) to stabilise the gas exchange. The applied procedure, despite its short sampling time and high responsiveness to drops in the light intensity, had some drawbacks. The method tended to overestimate stomata openness due to its slower adjustment to the actual light level. Despite this, we successfully applied it to evaluate the relationship between *g*_s_ and *T*_leaf_.

The values of the net photosynthetic rate (*P*_n_), stomatal conductance (*g*_s_), transpiration rate (*E*), and the abaxial leaf surface temperature (*T*_abaxial_) were recorded with a thermocouple. For the water-use efficiency (WUE) assessment, the intrinsic (*WUE*_int_; *P*_n_/*g*_s_) and instantaneous water-use efficiencies (*WUE*_ins_; *P*_n_/*E*) were estimated [55]. Each measurement comprised six replicates with four leaves per replicate. We analysed both control and DTT-infiltrated leaves to evaluate whether the DTT used for NPQ assessment influenced the leaf gas exchange parameters under constant PAR.

### 4.7. Models for Fitting of Experimental Data of T_leaf_ and NPQ or g_s_

The fitting of experimental data regarding leaf temperature change (Δ*T*) and the efficiency of light conversion to heat with the qZ component of non-photochemical quenching (ΔNPQ) or stomatal conductance (*g*_s_), NPQ induction, and gas exchange parameters was performed using OriginPro version 2023b (OriginLab Corporation, Northampton, MA, USA). The models for fitting the experimental data were applied as specified for each case and are reported with the adjusted R^2^ (R_a_^2^) value, used to determine the goodness of data fitting. In particular, we applied the following:Linear regression fitting model (1) and the double logarithmic reciprocal function of the Bradley regression model (nonlinear, (2)) for fitting data (*x*) regarding light-induced leaf temperature increase (Δ*T*) and the efficiency of light conversion to heat based on the qZ component of non-photochemical quenching (ΔNPQ) at 396 µmol m^−2^ s^−1^ B (450 nm) light:
(1)y=a+b×x
where *a* is an intercept and *b* is a slope.
(2)y=a×ln(−b×ln(x))
where *a* and *b* are regression coefficients.An allometric model (nonlinear, (3)) for fitting data (*x*) regarding the leaf abaxial surface temperature (*T*_abaxial_) and stomatal conductance (*g*_s_) at 400 µmol m^−2^ s^−1^ RGB (1:1:1) light:
(3)$$y=a\times x^{b}$$
where *a* is the coefficient of the equation and *b* is a power.A log-logistic equation with a variable Hill slope (*p*) model (nonlinear, (4)) for fitting data (*x*) regarding the leaf abaxial surface temperature (*T*_abaxial_) and stomatal conductance (*g*_s_) at 0–2000 µmol m^−2^ s^−1^ RB (R:B 10:1) light:
(4)$$y=A_{1\;}+\frac{A_{2}-A_{1}}{1+10^{{\left(Logx_{0}-x\right)}^{p}}}$$
where *A*_1_ is the bottom asymptote, *A*_2_ is the top asymptote, Log*x*_0_ is the centre, and *p* is the Hill slope.

Models for fitting the gas exchange parameter data against PAR: the asymptotic regression model ($y=a-b\times c^{x}$, where *a* is the asymptote, *b* is the response range, *c* is the rate), the lognormal cumulative distribution function ($y=y_{0}+A\;\int_{0}^{x}\frac{1}{\sqrt{2\pi }wt}e^{-\frac{{\left(\ln\left(t\right)-x_{c}\right)}^{2}}{2w^{2}}}dt$, where *y*_0_ is the offset, *A* is the amplitude, *x*_c_ is the log mean, and *w* is the standard deviation), the logistic regression model ($y=\frac{A_{1}-A_{2}}{1+{\left(\frac{x}{x_{0}}\right)}^{p}}+A_{2}$, where *A*_1_ is the initial value, *A*_2_ is the final value, *x*_0_ is the centre, and *p* is the power), and a sigmoidal model ($y=\frac{A_{1}-A_{2}}{1+e^{\left(x-x_{0}\right)/dx}}+A_{2}$, where *A*_1_ is the initial value, *A*_2_ is the final value, *x*_0_ is the centre, and *dx* is the constant).

### 4.8. Statistical Analysis

Statistical analyses were performed using Statistica 13.3 software (StatSoft Inc., Oklahoma, OK, USA). The normal distribution of variables was verified using the Shapiro–Wilk test, and the equality of variances was evaluated using Levene’s test. One-way ANOVA and post hoc Tukey’s HSD tests were employed to analyse the differences between the investigated groups. The data are presented as mean with standard deviation (±SD). Statistical significance was determined at the 0.05 level (*p* = 0.05).

## 5. Conclusions

Our results clearly document that under similar lighting and temperature conditions, the adaxial leaf surface of tomato plants irradiated with moderate light intensity (400 µmol m^−2^ s^−1^) presented significantly higher temperatures than the non-irradiated abaxial side. Thus, one might conclude that a major factor contributing to adaxial side temperature is irradiation exposure. However, an exception to this was G plants, which presented similar foliar temperatures on both leaf sides, thus putting into doubt the postulated influence of direct light exposure on foliar temperature. Additionally, we noticed discrepancies among the recorded *T*_leaf_ of plants under different spectra, as was revealed under moderate light intensity. We found that under constant PAR, the *T*_adaxial_ temperature was the highest for B-light-grown plants and the lowest for the G-light-grown plants. Taking these results into account, we postulate that a crucial factor contributing to the observed *T*_adaxial_ discrepancies among treatments was the spectrum-related NPQ amplitude characteristic, which was responsible for the uneven dissipation of absorbed excessive energy within the chloroplasts of the upper side of mesophyll cells. The basis of the observed NPQ difference among treatments was previously revealed via the distinct patterns of PsbS and VDE protein accumulation, i.e., increased levels in B-grown plants and reduced levels in plants grown under a monochromatic G or G-enriched mixed RGB spectrum. In this research, we also proved that DTT application, which inhibits the zeaxanthin-dependent NPQ component and significantly lowers its amplitude, was also responsible for a concomitant decrease in foliar temperature. As documented in previous studies, in the case of the influence of stomatal conductance on leaf energy balance, we indeed noted that *g*_s_ influences *T*_abaxial_. At the same time, however, this regulation occurs only with limited stomatal aperture and density; thus, under higher light intensities, the cooling effect fails to regulate *T*_leaf_.

## Figures and Tables

**Figure 1 ijms-24-16911-f001:**
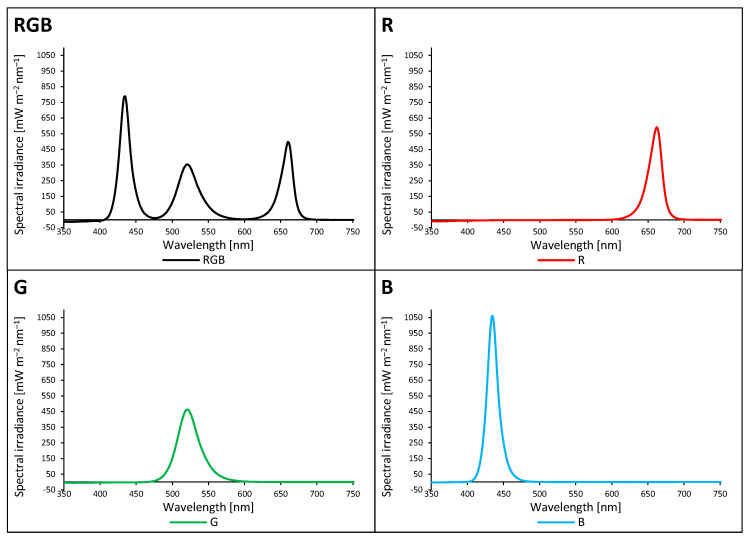
The light spectra of growth chambers were recorded with a spectroradiometer at six locations at the level of the apical bud and then averaged. Plants in the RGB (red–green–blue) chamber were grown under 180 µmol m^−2^ s^−1^, while the R, G, and B groups were grown under 80 µmol m^−2^ s^−1^. RGB states for the control plants (R:G:B = 1:1:1). The rest of the plants were grown solely under a monochromatic component of the RGB spectrum, provided via LED RhenacM12 lamps (PXM).

**Figure 2 ijms-24-16911-f002:**
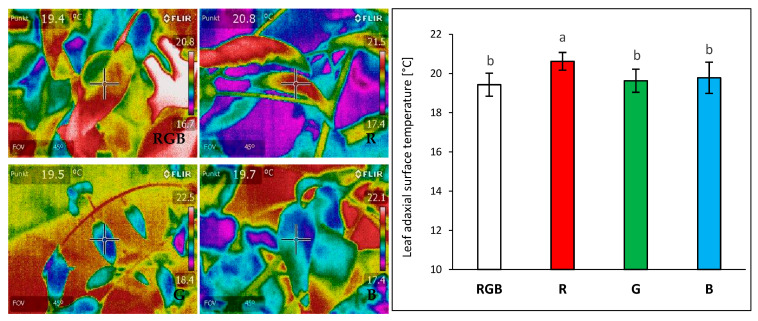
Effect of growth light quality on the adaxial surface temperature of tomato plant (*Solanum lycopersicum* L. cv. Malinowy Ozarowski) leaves 28 DAT, measured with a forward-looking infrared (FLIR) camera (FLIR E50) under chamber-specific light and air temperature (22 ± 1 °C) conditions. The distance from the thermal camera to the leaf surface was approximately 0.3 m. Each bar represents the average ± SD of six independent measurements (*n* = 6). Different letters (a, b) indicate significant differences between treatments at *p* = 0.05 with a Tukey’s HSD test.

**Figure 3 ijms-24-16911-f003:**
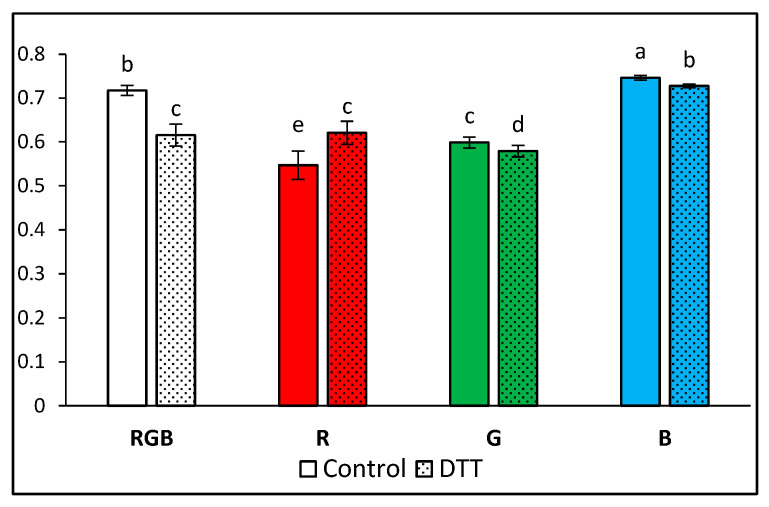
The maximal photochemical yield of PSII, Fv/Fm (relative unit: rel. unit), in water- (control) or 5 mM DTT-infiltrated (DTT) leaves of tomato plants (*Solanum lycopersicum* L. cv. Malinowy Ozarowski) 28 DAT, grown under different light conditions. Detached leaves were infiltrated for 1 h in the dark, followed by pre-illumination at 400 µmol m^−2^ s^−1^ of RGB light for the next 30 min. The minimal fluorescence level (Fo) was examined by measuring modulated blue light (450 nm, 0.01 µmol m^−2^ s^−1^). The maximal fluorescence level (Fm) was determined via a 0.8 s saturating blue light pulse (SP = 450 nm, 5000 µmol m^−2^ s^−1^). Each bar represents the average ± SD of six independent measurements (*n* = 6). Different letters (a–e) indicate significant differences between the treatments at *p* = 0.05 with a Tukey’s HSD test.

**Figure 4 ijms-24-16911-f004:**
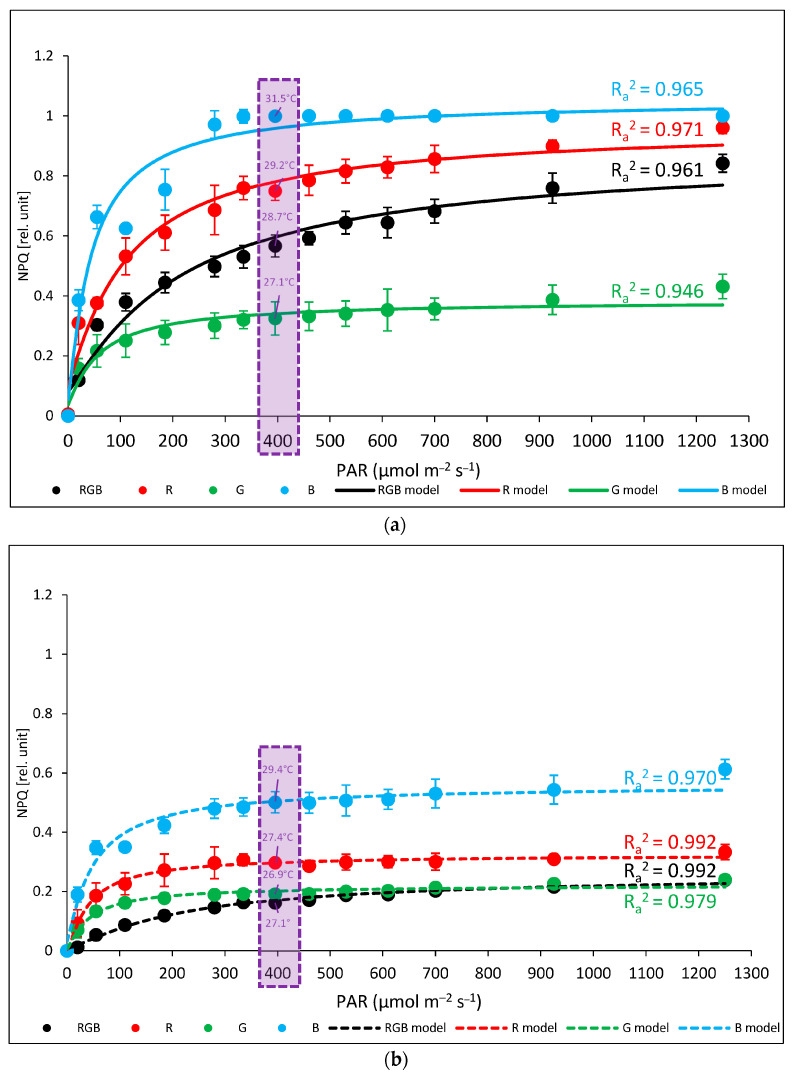
Non-photochemical quenching, NPQ (relative unit: rel. unit), in water- (control; **a**) or 5 mM DTT-infiltrated (DTT, **b**) leaves of tomato plants (*Solanum lycopersicum* L. cv. Malinowy Ozarowski) 28 DAT, grown under different light conditions. Detached leaves were infiltrated for 1 h in the dark, followed by pre-illumination at 400 µmol m^−2^ s^−1^ of RGB light for the next 30 min. The dynamics of the NPQ were determined in re-darkened (30 min) leaf samples illuminated for 5 min at the following steps: 0, 20, 55, 110, 185, 280, 335, 395, 460, 530, 610, 700, 925, and 1250 μmol m^−2^ s^−1^ of blue actinic light (AL = 450 nm) and a saturating blue light pulse (SP = 450 nm, 5000 µmol m^−2^ s^−1^, duration of 0.8 s) applied every 20 s at a given AL intensity. Each data point represents the average ± SD of six independent measurements (*n* = 6; black—RGB, red—R, green—G, blue—B chamber). The inverse exponential regression model (Exp3P1Md) was employed to fit the experimental data. Fitting was applied (as specified in the Section 4) and reported with an adjusted R^2^ (R_a_^2^) value to determine the goodness of data fitting. The adaxial surface temperature of leaves (purple dotted frame) was recorded simultaneously during chlorophyll fluorescence imaging (at 25 ± 1 °C) after the NPQ parameter had reached a plateau phase of induction at 396 µmol m^−2^ s^−1^, as determined using a forward-looking infrared (FLIR) camera (FLIR E50, distance 0.3 m).

**Figure 5 ijms-24-16911-f005:**
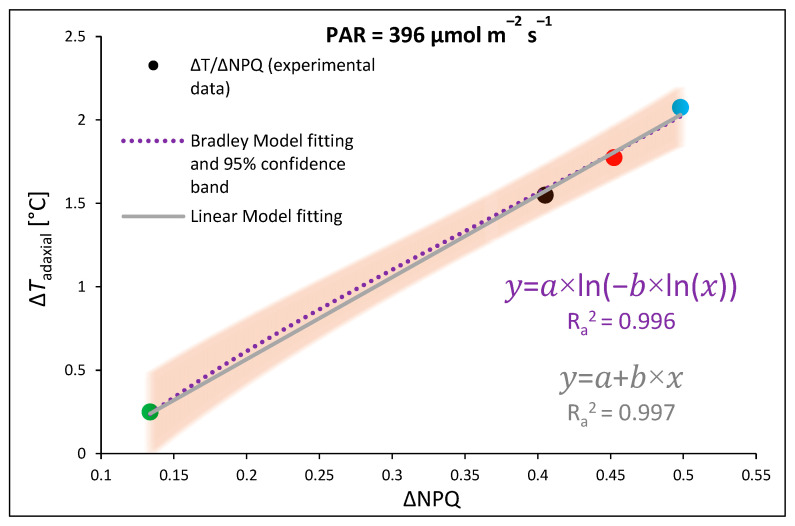
The relationship of light-induced leaf adaxial surface temperature increase (Δ*T*) and the efficiency of light conversion to heat with qZ component of non-photochemical quenching (ΔNPQ) at constant PAR. Experimentally obtained data (points; black—RGB, red—R, green—G, blue—B chamber) regarding the relationship between Δ*T* and ΔNPQ were assessed by subtracting the values of adaxial leaf surface temperature (FLIR) or NPQ (at 396 µmol m^−2^ s^−1^, 450 nm) estimated for the control and DTT-infiltrated leaves. The theoretical curve (dotted purple line) with a confidence band (95%, transparent orange area) and solid line (grey) were calculated based on the Bradley and linear regression models, respectively, employed to fit the experimental data of Δ*T*/ΔNPQ. Fitting was applied (as specified in the Section 4) and is reported with equations and adjusted R^2^ (R_a_^2^) values, which were used to determine the goodness of data fitting.

**Figure 6 ijms-24-16911-f006:**
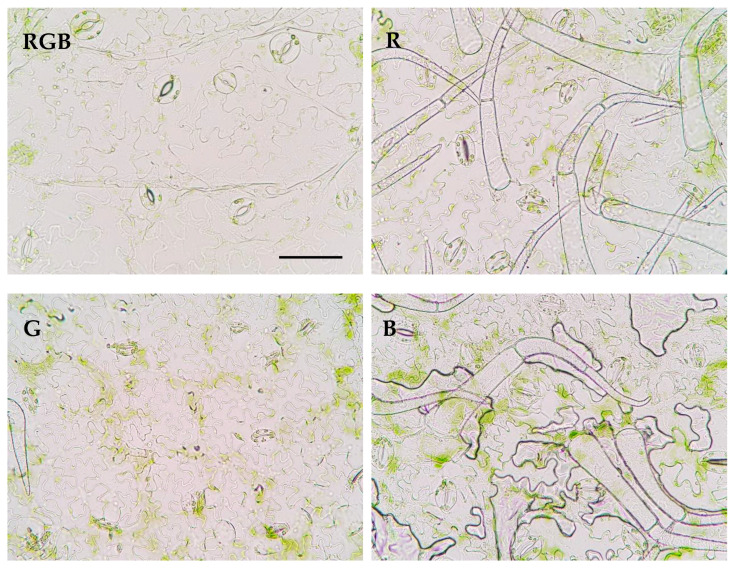
Effect of RGB, R, G, or B LED light on the distribution of the stomata in the abaxial epidermal layer of tomato plant (*Solanum lycopersicum* L. cv. Malinowy Ozarowski) leaves 28 DAT. Randomly selected images of leaf epidermises obtained by direct peeling taken under a Nikon Eclipse E100 light microscope (see the Section 4 for details), magnification of ×400. Bar = 50 µm.

**Figure 7 ijms-24-16911-f007:**
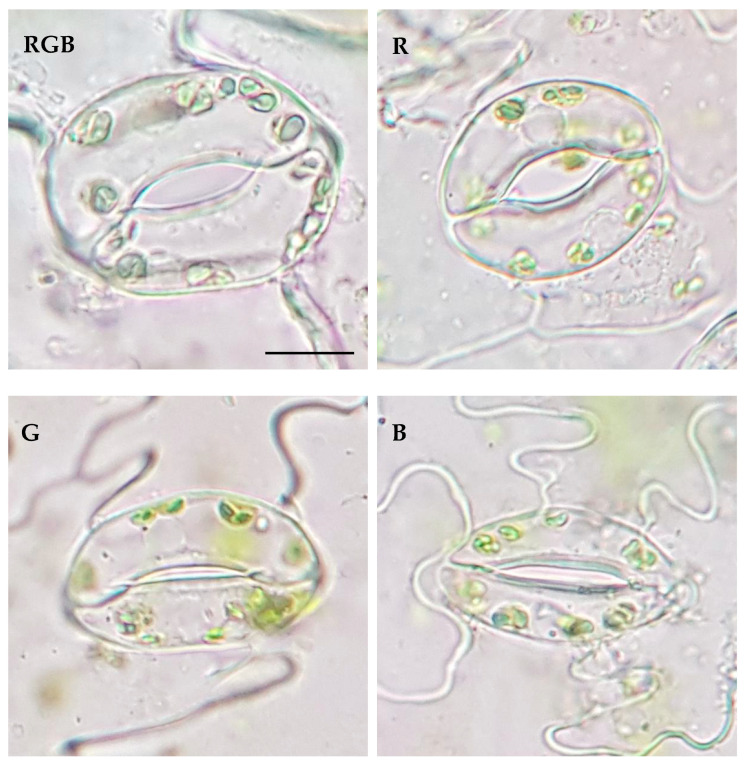
Effect of RGB, R, G, or B LED light on the anatomy of the stomata in the abaxial epidermal layer of tomato plant (*Solanum lycopersicum* L. cv. Malinowy Ozarowski) leaves 28 DAT. Randomly selected images of leaf epidermises obtained by direct peeling, taken using a Nikon Eclipse E100 light microscope (see the Section 4 for details), magnification of ×1000. Bar = 10 µm.

**Figure 8 ijms-24-16911-f008:**
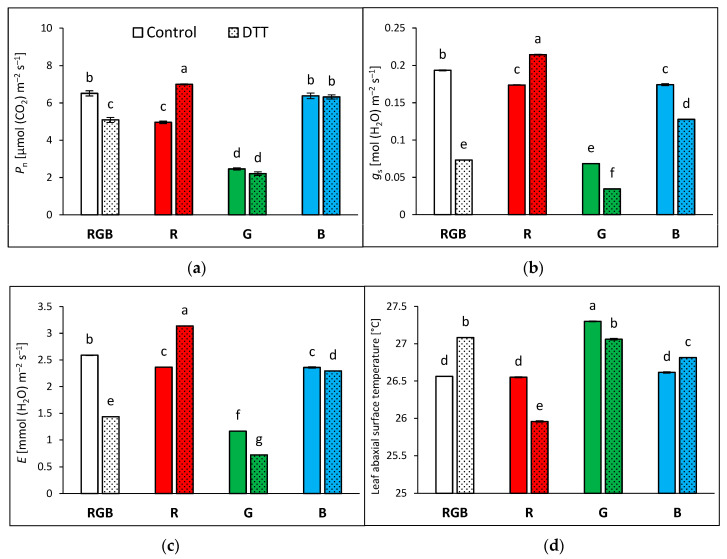
Gas exchange parameters of water- (control) or 5 mM DTT-infiltrated (DTT) leaves of tomato plants (*Solanum lycopersicum* L. cv. Malinowy Ozarowski) 28 DAT, measured in a 2 × 3 cm transparent chamber illuminated with 400 µmol m^−2^ s^−1^ of the RGB spectrum (R:G:B = 1:1:1). Detached leaves were infiltrated for 1 h in the dark, followed by pre-illumination at 400 µmol m^−2^ s^−1^ of RGB light for the next 30 min. (**a**) Net photosynthetic rate (*P*_n_), (**b**) stomatal conductance (*g*_s_), (**c**) transpiration rate (*E*), and (**d**) temperature of leaf abaxial surface. Each bar represents the average ± SD of six independent measurements (*n* = 6). Different letters (a–g) indicate significant differences between treatments at *p* = 0.05 with a Tukey’s HSD test.

**Figure 9 ijms-24-16911-f009:**
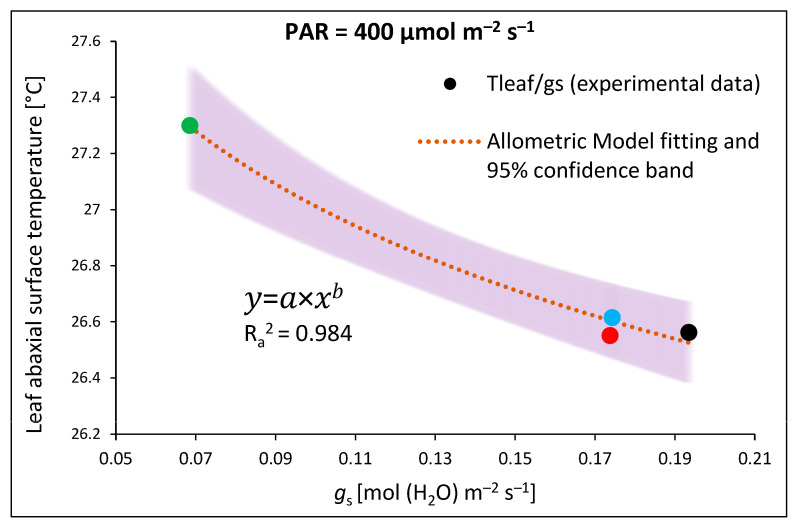
Relationship of leaf abaxial surface temperature (*T*_abaxial_) and stomatal conductance (*g*_s_) at constant PAR. Experimentally obtained data (points; black—RGB, red—R, green—G, blue—B chamber) regarding the relationship between *T*_abaxial_ and *g*_s_ were assessed with 400 µmol m^−2^ s^−1^ of the RGB spectrum (R:G:B = 1:1:1). The theoretical curve (dotted orange line) and confidence band (95%, transparent purple area) were calculated based on the Allometric model employed to fit the experimental data of *T*_abaxial_/*g*_s_. Fitting was applied (as specified in the Section 4) and is reported with an equation and an adjusted R^2^ (R_a_^2^) value, used to determine the goodness of data fitting.

**Figure 12 ijms-24-16911-f012:**
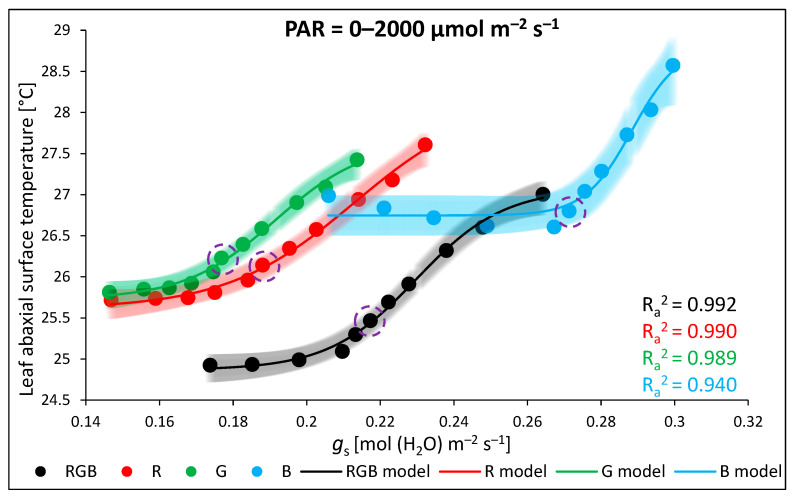
Relationship between leaf abaxial surface temperature (*T*_abaxial_) and stomatal conductance (*g*_s_) at increasing PAR. Experimentally obtained data (points; black = RGB, red = R, green = G, blue = B chamber) regarding the relationship between *T*_abaxial_ and *g*_s_ were assessed with 0–2000 µmol m^−2^ s^−1^ of RB light (R:B 10:1). The threshold level of the limitation of the *g*_s_-related cooling effect is depicted with data points (enclosed in purple circles) at 400 µmol m^−2^ s^−1^ of PAR. The theoretical curves (solid line) and confidence bands (95%, transparent area) were calculated based on the log-logistic equation with a variable Hill slope model employed to fit the experimental data of *T*_abaxial_/*g*_s_. Fitting was applied (as specified in the Section 4) and is reported with an adjusted R^2^ (R_a_^2^) value, used to determine the goodness of data fitting.

**Table 1 ijms-24-16911-t001:** Stomatal traits of the abaxial leaf surface of tomato plants (*Solanum lycopersicum* L. cv. Malinowy Ozarowski) 28 DAT, grown under different light conditions.

Parameter	Treatment			
Stomatal Traits (Abaxial Leaf Surface)	RGB	R	G	B
Stomatal complex width (µm)	24.00 ± 3.49 a	19.39 ± 1.87 b	17.84 ± 0.81 b	17.62 ± 2.14 b
Stomatal complex length (µm)	29.82 ± 3.19 a	24.24 ± 2.17 b	25.06 ± 3.42 b	26.07 ± 1.86 b
Stomatal width/length ratio	0.80 ± 0.06 a	0.80 ± 0.04 a	0.72 ± 0.1 b	0.68 ± 0.08 b
Stomatal complex area (µm^2^)	568.44 ± 139.38 a	373.63 ± 59.72 b	349.19 ± 61.03 b	347.07 ± 60.38 b
Pore width (µm)	4.04 ± 0.88 ab	4.34 ± 0.91 a	1.90 ± 0.67 c	3.65 ± 0.99 b
Pore length (µm)	12.32 ± 1.59 b	12.02 ± 1.22 b	12.26 ± 2.18 b	13.78 ± 1.62 a
Aspect ratio of pores (width/length)	0.33 ± 0.06 a	0.36 ± 0.07 a	0.16 ± 0.05 c	0.26 ± 0.06 b
Stomatal pore area (µm^2^)	34.14 ± 9.12 a	35.27 ± 8.69 a	17.15 ± 6.00 b	29.90 ± 8.89 a
Stomatal density *S*_d_ (no. mm^−2^)	117.22 ± 5.08 b	140.66 ± 8.79 a	90.84 ± 13.43 c	137.73 ± 18.30 a
Pore area per leaf area (µm^2^ mm^−2^)	4002.09 ± 1069.27 b	4960.61 ± 1222.53 a	1557.91 ± 545.44 c	4100.42 ± 1219.19 b
Total pore area per total leaf surface (µm^2^ cm^−2^)	18,019.50 ± 4814.41 a	11,952.84 ± 2945.76 b	4321.61 ± 1513.05 c	19,751.74 ± 5872.84 a

Values are means of ten replicates ± SD. For the individual stomatal traits, a magnification of ×1000 was used or ×400 for *S*_d_ and pore area per leaf area/total pore area per total leaf surface. Different letters (a–c) in the same row indicate significant differences between the treatments at *p* = 0.05 with a Tukey’s HSD test.

## Data Availability

The data presented in this study are available on request from the corresponding author. The data are not publicly available due to the strict management of various data and technical resources within the research teams.

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
