# Peer review of "Growth Light Quality Influences Leaf Surface Temperature by Regulating the Rate of Non-Photochemical Quenching Thermal Dissipation and Stomatal Conductance"

_ijms, 2023, doi:10.3390/ijms242316911_

Round 1

Reviewer 1 Report

Comments and Suggestions for Authors

Reviewer 2 Report

Comments and Suggestions for Authors

The manuscript covers the effects of light on photosynthetic parameters, it is already known from extensive global research on photophysiology that monochromatic and mixed color LED lamps have different effects on the functioning of the photosynthetic apparatus. The novelty of this study is that NPQ, stomatal conductance is discussed together with leaf surface temperature. Although the article is quite good, the results section includes not only their discussion but also a discussion of what should be separated according to the structure of the manuscript. All citations and discussion sentences that try to explain why such results were obtained must be removed from the results section, all this information must be presented in the discussion section. I do not provide specific lines, because almost the entire part of the results should be marked. 

Reviewer 3 Report

Comments and Suggestions for Authors

The manuscript with the title “Growth light quality influences the temperature of the leaf surface via regulation of the rate of NPQ thermal dissipation and stomatal conductance” studied the influence of monochromatic LED lamps and mixed RGB spectrum on leaf temperature on abaxial and adaxial surfaces of tomato.

Specific comments

Introduction - In addition to the Lines 72-73, I would suggest to also mention whether differences between the type of mesophyll in plants was shown to play an influence (in previous published literature).

The fact that temperature for abaxial surface corresponds to gs could in any way be in relation to the fact that tomato plant leaf has more abundant stomata on lower surface than upper surface?

Material and Method subchapter 4.5. explains that only abaxial stomata parameters were conducted, therefore I suggest to mention this aspect at Table 1 from Results.

Figures 9 and 10, vertical axes are superimposed.

Best regards.
